# Activating Hippo Pathway via Rassf1 by Ursolic Acid Suppresses the Tumorigenesis of Gastric Cancer

**DOI:** 10.3390/ijms20194709

**Published:** 2019-09-23

**Authors:** Seong-Hun Kim, Hua Jin, Ruo Yu Meng, Da-Yeah Kim, Yu Chuan Liu, Ok Hee Chai, Byung Hyun Park, Soo Mi Kim

**Affiliations:** 1Department of Internal Medicine, Chonbuk National University Medical School, Jeonju 54907, Korea; shkimgi@jbnu.ac.kr; 2Department of Physiology, Chonbuk National University Medical School, Jeonju 54907, Korea; jinhuaxy@126.com (H.J.); kathymeng1216@gmail.com (R.Y.M.); kdyeah@jbnu.ac.kr (D.-Y.K.); liu_yuchuan@126.com (Y.C.L.); 3Department of Anatomy and Institute for Medical Sciences, Chonbuk National University Medical School, Jeonju 54907, Korea; okchai1004@jbnu.ac.kr; 4Department of Biochemistry, Chonbuk National University Medical School, Jeonju 54907, Korea; bhpark@jbnu.ac.kr; 5Research Institute of Clinical Medicine of Chonbuk National University, Jeonju 54907, Korea; 6Biomedical Research Institute of Chonbuk National University Hospital, Jeonju 54907, Korea

**Keywords:** ursolic acid, Hippo signaling, gastric cancer cells, proliferation, metastasis

## Abstract

The Hippo pathway is often dysregulated in many carcinomas, which results in various stages of tumor progression. Ursolic acid (UA), a natural compound that exists in many herbal plants, is known to obstruct cancer progression and exerts anti-carcinogenic effect on a number of human cancers. In this study, we aimed to examine the biological mechanisms of action of UA through the Hippo pathway in gastric cancer cells. MTT assay showed a decreased viability of gastric cancer cells after treatment with UA. Following treatment with UA, colony numbers and the sizes of gastric cancer cells were significantly diminished and apoptosis was observed in SNU484 and SNU638 cells. The invasion and migration rates of gastric cancer cells were suppressed by UA in a dose-dependent manner. To further determine the gene expression patterns that are related to the effects of UA, a microarray analysis was performed. Gene ontology analysis revealed that several genes, such as the Hippo pathway upstream target gene, ras association domain family (*RASSF1*), and its downstream target genes (MST1, MST2, and LATS1) were significantly upregulated by UA, while the expression of YAP1 gene, together with oncogenes (FOXM1, KRAS, and BATF), were significantly decreased. Similar to the gene expression profiling results, the protein levels of RASSF1, MST1, MST2, LATS1, and p-YAP were increased, whereas those of CTGF were decreased by UA in gastric cancer cells. The p-YAP expression induced in gastric cancer cells by UA was reversed with RASSF1 silencing. In addition, the protein levels in the Hippo pathway were increased in the UA-treated xenograft tumor tissues as compared with that in the control tumor tissues; thus, UA significantly inhibited the tumorigenesis of gastric cancer in vivo in xenograft animals. Collectively, UA diminishes the proliferation and metastasis of gastric cancer via the regulation of Hippo pathway through Rassf1, which suggests that UA can be used as a potential chemopreventive and therapeutic agent for gastric cancer.

## 1. Introduction

Gastric carcinoma is one of the most deadly cancers and it is associated with a high mortality rate [1,2,3,4]. The estimated new case of gastric cancer is about 950,000 and the estimated death of gastric cancer accounts for about 800,000 deaths worldwide every year [3,4]. The incidence of gastric cancer in Asia remains high, including in South Korea, [3,4], but a continuous diminution in gastric cancer occurrence has been reported in the developed countries of Europe and in America since the middle of the 20^th^ century [3,4]. Although the attack rate of gastric cancer has constantly diminished over the last century, this cancer remains a critical cause of morbidity and mortality globally due to its unclear pathogenesis. Even though surgery, chemotherapy, radiation therapy, and immunotherapy are used for the amelioration of gastric cancer, these therapies are expensive and inefficient. Therefore, it is imperative to find a novel effective remedy to obtain improved clinical results in patients with gastric cancer.

Ursolic acid (UA) is a natural compound that is extracted from multifarious medicinal plants, including herbal plants. Studies have revealed that UA has anti-inflammatory effects and induces apoptosis in breast, pancreatic, colon, prostate, and lung cancers; melanoma; and, leukemia [5,6,7,8,9,10]. UA has been shown to suppress tumorigenesis [11,12] and antiangiogenic activity [13] to regulate multiple cancer-related signaling mechanisms, including cell cycle regulation [14,15], activation of caspases [16], inhibition of anti-apoptotic genes [17,18], COX-2 [19], matrix metallopeptidase-9 [20], and tyrosine kinase [21]. Some recent reports revealed that UA has antitumor effects in gastric cancer by inhibiting various carcinogenic processes [22,23]. Xiang et al. reported that UA inhibits the proliferation of gastric cancer cells by targeting miR-133a [22]. Kim et al. showed that UA inhibits the invasion of gastric cancer cells by decreasing the expression levels of MMP-2 [23]. However, the precise biological function of UA, and what specific mechanism that UA has antitumor potential against gastric cancer is not fully investigated.

The Hippo pathway has been acknowledged in Drosophila and it is evolutionarily preserved in mammals as Mst1/2, WW45, LATS1/2, Yap, and Mob1 to regulate cell survival and growth [24,25,26,27]. This pathway has come under special scrutiny because this pathway is crucial in control cell fate with proliferation, stimulating speculation that many members of this pathway are involved in tumorigenesis [27]. When Hippo signaling is activated, phosphorylated Yap is displaced in the cytoplasm, which encourages degradation, whereas when Hippo signaling is inactivated, unphosphorylated Yap transfers to the nucleus and induces the transcriptional activity of genes that are involved in cell growth [27,28,29]. Although the Hippo pathway plays an important role in various cancers, no studies have been reported on the association of UA with the Hippo pathway. Hence, clarifying the relationship between UA and the signaling regulation of the Hippo pathway appears to be necessary in human cancers, such as gastric cancer.

Gene expression profiling of cancer has been studied to identify gene expression patterns that are associated with small molecule’s effect on cancer cells, elucidating specific biological molecular pathway of antitumor effect of small molecule on cancer cells [30,31,32]. In the present study, the gene expression patterns that are related to the effects of UA were determined to address the mechanistic studies of UA in gastric cancer cells. Gene ontology analysis revealed that several genes, including the upstream regulator of the Hippo signaling pathway (MST1, MST2, LATS1) and ras association domain family (RASSF1), were significantly upregulated and its downstream target genes of Hippo signaling pathway (YAP1, CTFG, FOXM1, KRAS, BATF) were significantly downregulated by UA treatment. In addition, UA suppressed the proliferation and metastasis of gastric cancer cells and inhibited the tumorigenesis of gastric cancer in vivo in xenograft animals through the activation of the Hippo pathway. Therefore, our data suggest that UA obstructs the proliferation and metastasis of gastric cancer by regulating the Hippo pathway.

## 2. Results

### 2.1. Cell Proliferation Inhibition by UA in Gastric Cancer Cells

We performed the MTT assay on SNU484 and SNU638 cell line to investigate the effect of UA on gastric cancer cells. Our results showed that the growth of the SNU484 and SNU638 cells was suppressed in a dose-dependent manner after UA treatment at 72 h. UA (50–100 μΜ) led to a significant growth suppression of SNU484 cells (80–95% at 72 h) and SNU638 cells (50–90% at 72 h) as compared with controls, which indicates a significant inhibitory effect of UA on gastric cancer cells (Figure 1). These data suggest that UA significantly inhibited cell growth, and higher concentration UA treatment could induce greater effect of cell death.

### 2.2. Inhibitory Effect of UA on Colony Formation in Gastric Cancer Cells

We next performed the colony formation assay to further investigate the antitumor effect of UA in gastric cancer cells. As shown in Figure 2, the size and number of colonies were significantly suppressed in a dose-dependent manner after the treatment of SNU484 and SNU638 cells with UA, which suggests that UA significantly inhibited the growth of gastric cancer cells.

### 2.3. Induction of Apoptosis by UA in Gastric Cancer Cells

We further determined the levels of apoptotic proteins: PARP, caspase9, and caspase3 to clarify the effects of UA on apoptosis in gastric cancer cells. Our results showed that UA increased cleaved-PARP and cleaved-caspase 9 protein levels in a dose-dependent manner (0, 10, 25, 50 μM), whereas the expression levels of PARP, caspase 9, and caspase-3 were significantly decreased in a dose-dependent manner (0, 10, 25, 50 μM) by UA in SNU484 and SNU638 cells (Figure 3A). We further tested the apoptotic effect of UA in gastric cancer cells via flow cytometry analysis. Annexin V/PI staining revealed that UA (25–50 μM) significantly induced apoptosis in SNU484 and SNU638 cells (Figure 3B). Flow cytometry was used to determine the relative cell figures in each cell cycle phase. As shown in Figure 3C, the sub-G1 phase, which was considered apoptotic cells, increased in SNU484 cells after treatment with UA at 50 μM and in SNU638 cells after treatment with UA at 10, 25, and 50 μM. These results indicate that UA led to the apoptosis of gastric cancer cells.

### 2.4. Inhibitory Effect of UA on Migration in Gastric Cancer Cells

We performed wound healing assay to test whether UA regulates gastric cancer cell migration. As shown in Figure 4, the migration rates of SNU484 and SNU638 cells were markedly reduced in a dose-dependent manner at 24 h and 48 h. These data show that UA diminishes the migration ability of gastric cancer cells.

### 2.5. Inhibitory Effect of UA on Invasiveness in Gastric Cancer Cells

A Matrigel invasion assay was performed to verify the effect of UA on gastric cancer cell invasion. As shown in Figure 5, UA significantly repressed the invasion of the SNU484 and SNU638 cells. These results indicate that UA suppresses the invasion abilities of gastric cancer cells.

### 2.6. Gene Expression of the Hippo Pathway by UA in Gastric Cancer Cells

Transcriptome analysis was performed using microarray to further explore the effect of UA on genome-wide expression levels in gastric cancer cells. UA significantly increased the expression of RASSF1 and Mst1 genes and mildly increased the Mst2 and LATS1 (Figure 6). The expression of YAP1 gene together with oncogenes (FOXM1, KRAS, and BATF) was significantly diminished in SNU484 cells by UA treatment.

### 2.7. Activation of Hippo Signaling Pathway by UA in Gastric Cancer Cells

We determined the protein levels of the Mst1, Mst2, p-Mob1, and LATS1 to further examine the effect of UA on the Hippo pathway. As shown in Figure 7A, UA increased the expression of the Hippo signaling pathway proteins (Mst1, Mst2, p-Mob1, and LATS1) in a dose-dependent manner in SNU484 and SNU638 cells (0, 25, 50 μΜ). UA also significantly induced the protein expression levels of RASSF1 in a dose-dependent manner in SNU484 and SNU638 cells. In addition, the expression of phosphorylated Yap, an inactivate form of Yap, was also significantly increased by UA treatment, while the expression of Yap was not changed (Figure 7B). The protein expression of CTGF, a downstream target gene of YAP, was inhibited by UA in a dose-dependent manner in SNU484 and SNU638 cells (0, 25, and 50 μΜ; Figure 7B). We further tested whether UA treatment could facilitate RASSF1 using a silencing experiment with RNA interference in SNU484 and SNU638 cells because UA suppresses gastric cancer cell proliferation by regulating the Hippo signaling pathway. The expression of p-YAP was increased, whereas that of YAP was decreased after UA treatment in the SNU484 and SNU638 cells. The increased p-YAP expression induced in SNU484 and SNU638 cells by UA was reversed with RASSF1 silencing (Figure 7C). To further confirm the direct link between Hippo pathway with the cellular phenotypes, such as cell viabilities, we performed an experiment to determine the cell survival after UA treatment Yap siRNA-transfected SNU484 and SNU638 cells. As shown in Figure 7D, the cell survival was significantly decreased by YAP silencing. Furthermore, UA further suppressed the cell survivals reduced by Yap silencing in SNU484 and SNU638 cells (Figure 7D). These results suggest that there is a direct link showing that the Hippo pathway is responsible for the cellular phenotype cause by UA treatment. In agreement with increased protein levels by UA, the relative mRNA expression of RASSF1 was significantly increased by UA in a dose-dependent manner (0, 25, and 50 μΜ) in SNU484 and SNU638 cells (Figure 7E). Conversely, the relative mRNA expressions of YAP and CTGF were significantly decreased by UA in a dose-dependent manner in SNU484 and SNU638 cells (Figure 7E). Therefore, our results suggested that UA inhibits the gastric cancer tumorigenesis mediated through the regulation of Hippo signaling pathway.

### 2.8. Inhibitory effect of UA on Tumor Growth in Xenograft Animal

We further tested whether the antitumor effect of UA will be reflected in an in vivo xenograft animal model. We injected SNU484 cells into the right flank of SPF/VAF immunodeficient mice. Xenograft animals were divided into two groups and were treated daily in a five days subcutaneously (s.c.) with either control or UA (10 mg/kg) for 30 days. The body weight and tumor volume of mice were measured once every three days while using calipers. UA treatment resulted in a great suppression of tumor growth in SNU484 xenograft animals (Figure 8A,B). To further test the antitumor efficacy of UA *in vivo*, paraffin-embedded tissues were sectioned at 4 μM and stained with hematoxylin and eosin (H&E). As shown in Figure 8D, the control group showed a disseminated proliferation of small- to intermediated-sized cells; the cells exhibited varying shapes of hyperchromatic nuclei and scant cytoplasm. Meanwhile, minimal necrosis and moderate lymphocyte infiltration around the tumor cells were identified in the UA treatment group (Figure 8D). Taken together, these results suggest that UA significantly suppresses SNU484 xenograft tumor growth in vivo and prevents tumorigenesis.

### 2.9. Activation of the Hippo Signaling Pathway by UA in the Tumor Tissues of Xenograft Animals

We measured the protein expression levels of Hippo pathway proteins (Rassf-1, Mst1, Mst2, Sav1, Mob1, p-Mob1, YAP, p-YAP, and CTGF) and metastasis proteins (E-cadherin, MMP-9, Twist, and Vimentin) in the control and UA-treated xenograft tumor tissues to further evaluate whether UA regulates the tumorigenesis of gastric cancer in vivo through activating the Hippo pathway. The protein levels of Rassf1, Mst1, Mst2, Sav1, Mob1, and p-Mob1 increased in the UA-treated xenograft tumor tissues as compared with those in the control tumor tissues (Figure 9A). The protein levels of YAP and CTGF were decreased in the UA-treated xenograft tumor tissues when compared with those in the control tumor tissues (Figure 9B). In addition, the E-cadherin protein level was increased, whereas the MMP9, Twist, and Vimentin protein levels were decreased in the UA-treated xenograft tumor tissues as compared with those in the control tumor tissues (Figure 9C). Therefore, our results indicate that UA significantly suppresses SNU484 xenograft tumor growth and metastasis in vivo through the activation of the Hippo pathway.

## 3. Discussion

Cancer progress is a multifaceted procedure that is orchestrated by the communication among numerous signaling pathways [33]. Gastric cancer development is a multi-step process that accumulates various genetic alterations by the stimulation of oncogenes or the suppression of tumor suppressor genes [33]. The objective of this research was to discover the anti-proliferative effect of phytochemical UA on gastric cancer cells while using gene expression profiling. The main result is the observation that activating of the Hippo signaling pathway by UA suppresses the growth and invasion/migration of human gastric cancer cells.

The Hippo signaling pathway has gained pronounced attention to the scientists, because it plays a crucial role in regulating tissue growth all the way through their individual functions and other biological processes, including cell fate determination, mitosis, and pluripotency [34,35]. Up to now, the functions of the Hippo signaling pathway are found to control the size of organs, cell viability, the invasion/metastasis, and to maintain stem cell self- renewal and adaptability [36]. In recent days, the inactivation or abnormal regulation of the Hippo signaling pathway has been found in several cancers, including lung, esophagus, stomach, liver, ovary, and bladder [37,38,39,40,41]. This abnormal regulation of the Hippo signaling pathway might induce abandoned phenotypic variations, which results in cancer [42,43]. In gastric cancer, the Hippo pathway has been found to be dysregulated. Xu et al. reported that inactivation of Hippo promotes gastric tumorigenesis and cancer development [43]. Silencing of Hippo pathway components is frequently identified in gastric cancer and elevated expression of YAP, a Hippo pathway effector, is observed in gastric cancer [44]. The overexpression of YAP is significantly shortened overall survival in gastric cancer patients with strongly correlated with lymphatic metastasis [45,46,47]. Moreover, the knockdown of YAP inhibits proliferation, colony formation, and metastasis in various gastric cancer cell models in xenograft models [45,48,49], which suggests that the Hippo pathway plays an important role in gastric cancer and its potential targeting for gastric cancer. In the present study, UA drastically increased the activation of Hippo pathway components, such as RASSF1, Mst1, Mst2, and LATS1, by gene expression profiling study. Additionally, the expression of YAP1, FOXM1, KRAS, and BATF genes were decreased in gastric cancer cells by treatment of UA. In agreement with the gene expression profiling study, we found that UA increased the expression of the Hippo signaling pathway proteins Mst1, Mst2, p-Mob1, and LATS1 in gastric cancer cells. UA significantly induced the protein expression levels of RASSF1 and the expression of phosphorylated Yap, an inactive form of Yap, in a dose-dependent manner in gastric cancer cells. In addition, silencing RASSF1 significantly inhibited the expression of p-YAP in SNU484 and SNU638 cells. Moreover, silencing RASSF1 in addition to UA treatment in SNU484 and SNU638 cells significantly reduced the increased expression of p-YAP in these cells. Taken together, our results suggest that UA activates Hippo signaling pathway in gastric cancer cells.

Despite substantial efforts being made to develop the treatment of gastric cancer, the prognosis of gastric cancer patients is still underprivileged. Malignant proliferation, wide-ranging invasion, and lymphatic metastasis have been a key challenges that limit effective therapeutic strategies. Therefore, prevention or identifying potential therapeutic targets of gastric cancer is critical for improving outcomes. In fact, natural compounds are thriving recognized as chemopreventive agents. UA is a natural compounds derived from a large variety of medicinal plants, including apple peel. Studies have shown that UA has inhibitory effects on cell proliferation, tumorigenesis, angiogenesis, metastasis and induce apoptosis in various cancer types [5,6,7,8,9,10,11,12]. In gastric cancer, UA has been found to have antitumor effects by inhibiting various carcinogenic processes, including targeting miR-133a, decreasing the invasion ability, enhancing radiation effects, induce apoptosis, and the inhibition of COX-2 [22,23,50,51,52,53,54,55]. Our observations are in agreement with those made in earlier studies wherein antitumor effect was induced by UA treatment in gastric cancer cells. In the present study, we found that UA significantly inhibited cell growth and induced apoptosis via increasing the protein levels of cleaved-PARP and cleaved-caspase 9 in gastric cancer cells. We further confirmed that UA induces apoptosis through Annecxin V/PI staining and increased levels of sub-G1 percentages. Moreover, UA suppressed the migration and invasion of SNU484 and SNU638 cells. These results suggest that UA induced apoptosis via the activation of caspase 9 and suppressed migration and metastasis in gastric cancer cells. In an in vivo experiment, UA treatment significantly suppressed tumor growth and caused histological changes, such as minimal necrosis and moderate lymphocyte infiltration in SNU484 xenograft animals. In addition, the protein levels of Rassf1, Mst1, Mst2, Sav1, Mob1, and p-Mob1 were increased, whereas those of YAP and CTGF were diminished in UA-treated xenograft tumor tissues. The levels of metastasis-related proteins, such as MMP9, Twist, and Vimentin, were decreased, whereas those of E-cadherin protein were increased in UA-treated xenograft tumor tissues when compared with those in the control tumor tissues, which indicates that UA has significantly suppressed SNU484 xenograft tumor growth in vivo mediated by the activation of Hippo pathway. Therefore, UA-mediated apoptosis in gastric cancer cells appeared to be accompanied by a downregulation of YAP1 and the activation of Hippo pathway; thus, our results demonstrated a direct connection between apoptosis by UA treatment and activation of Hippo pathway, which suppressed the expression of YAP proteins, a downstream target gene of Hippo pathway in gastric cancer cells.

Taken together, the current study for the first time provides strong evidence that the inactivation of Yap and the activation of Hippo pathway components indicated that UA leads to the activation of the Hippo pathway, which might ultimately induce apoptosis and inhibit proliferation and invasion/metastasis of SNU484 and SNU638 cells and tumor growth in xenograft animals. Furthermore, UA markedly inhibits gastric cancer tumorigenesis through the stimulation of Hippo signaling pathway, which suppresses the expression of YAP proteins, suggesting that UA can be used as a potential chemopreventive and therapeutic agent for gastric cancer.

## 4. Materials and Methods

### 4.1. Cell Culture and Reagents

The SNU484 and SNU638 cell lines were obtained from the Korean Cell Line Bank and they were incubated in RPMI-1640 medium supplemented with 10% fetal bovine serum and 1% penicillin in 100-mm dishes under normal conditions at 37 °C with a 5% CO_2_-moistened environment. The following antibodies were obtained from the following commercial sources: Antibodies to PARP, caspase-9, caspase-3, Mst1, Mst2, Mob1, p-Mob-1, LATS1, YAP, p-YAP, Sav1, CTGF, MMP9, E-cadherin, Vimentin, and GAPDH were purchased from Cell Signaling Technology (Beverly, MA, USA). Antibodies to Rassf1 and CTGF were obtained from Santa Cruz Biotechnology (Santa Cruz, CA, USA). UA was obtained from Cayman chemical company (Ann Arbor, Michigan, USA).

### 4.2. Cell Proliferation Assay

The SNU484 and SNU638 cells were plated in 96-well plates and incubated for 24 h. Subsequently, the cells were treated with UA at the indicated concentrations (0, 10, 25, 50, 75, and 100 μM) for 48 h. Cell viability was measured while using 3-(4, 5-dimethylthiazol-2-yl)-2, 5-diphenyltetrazolium bromide (MTT) assays. 50 μL of MTT was added to the culture medium of growing cells at 48 h and the cells were further cultured at 37 °C for 4 h. After incubation, remove all the MTT solution and culture medium, add 200 μL of DMSO to the well, measure the absorbance at 540 nM while using a model Epoch microplate reader (BioTek, Winooski, VT, USA). This assay was performed in triple times.

### 4.3. Western Blot Analysis

Briefly, SNU484 and SNU638 cells were seeded and plated to attach for 24 h. UA was added to cell cultures at 0, 10, 25, or 50 µM concentrations and incubated for 72 h. Subsequently, the samples were collected and adjourned in a lysis buffer (Intron Biotechnology, Korea). Cells were extracted and incubated on ice for 10 min. The samples were then centrifuged at 13,200 rpm for 20 min. at 4 °C and the protein concentration was measured (BSA protein assay kit, Pierce Biotechnology, Inc., Rockford, IL, USA). The lysate was resolved on a gel and then moved to the membranes (PVDF, Bio-Rad, Hercules, CA, USA). The specific primary antibodies were displayed and the secondary antibodies were probed to the membranes. Protein bands were visualized while using a chemiluminescence kit (Amersham, Arlington Heights, IL, USA). The following antibodies were used: cleaved-PARP, PARP, cleaved-caspase-9, caspase-9, caspase-3, Rassf1, Mst1, Mst2, Mob1, p-Mob1, LATS1, YAP, p-YAP, CTGF, Sav1, E-cadherin, MMP-9, Vimentin, Twist, and GAPDH. All of the antibodies were diluted to 1:1000. Quantification analysis of Western blotting band was performed while using Image J.

### 4.4. Soft Agar Colony Formation

The bottom layer of soft agar (1%) was prepared in a six-well plate, and the top layer (0.7%) was prepared with 5 × 10^4^ cells /well in a single cell suspension. The cells were divided into groups needed. The cells were cultured in an incubator with 5% CO_2_ at 37 °C for at least two weeks and observed for colony formation by microscopy. Colonies of >30 cells were counted and the experiments were repeated in triplicate.

### 4.5. Matrigel Invasion Assay

In vitro cell invasion was performed while using the BD BioCoat^TM^ Matrigel^TM^ Invasion Chamber (BD Biosciences, San Jose, CA, USA). Briefly, the Matrigel-coated chambers were rehydrated in a humidified tissue culture incubator at 37 °C with 5% CO_2_. Cells (2.5 × 10^4^) were seeded in 500 µL medium in each Matrigel-coated transwell insert and the lower chamber of the transwell was filled with 500 µL medium. After incubation, the inserts were washed and stained with a Diff-Quik kit (Sysmex Corp., Kobe, Japan), sequentially transfer the inserts through each solution. Observe and photograph the invading cells under microscope at approximately 40–200× magnifications, depending on cell density. Count cells in five fields and calculate the invasion rates.

### 4.6. Wound Healing Assay

A wound healing assay was performed to determine the effect of UA on the migration ability in the SNU484 and SNU638 cell lines. A wound was made through the monolayer using a 200 µL pipette tip. Wounds were measured over a time course to calculate the migration rate. The experiments were performed more than three times.

### 4.7. Microarray Experiment and Data Analysis

Gene expression profiling was performed according to the manufacturer’s protocols (Illumina Inc, CA, USA). Briefly, total RNA was extracted, labeled, and hybridized with cyanin-3-streptavidin (GE Healthcare, NJ, USA). The chips were scanned with the illumine BeadArray Reader (Ilumina, CA, USA) and data were extracted while using Genome Studio (Illimina, CA, USA). Using quantile normalization, the data were normalized and transformed into a log 2 base while using BRB array tools (https://brb.nci.nih.gov). Cluster 3.0 and Treeview programs were used for generating the heat map of gene expression [56].

### 4.8. Cell Cycle Analysis

The SNU484 and SNU638 cells were seeded into 60-mm dishes at a concentration of 10^6^ cells in RPMI 1640. The cells were treated with UA (0, 10, 25, or 50 µM) for 48 h and then fixed with 75% ethanol for 2 h at −20 °C. After fixation, the cells were stained with propidium iodide (Sigma Chemicals, St. Louis, MO, USA) at 37 °C for 30 min. An FACStar flow cytometer (Becton-Dickinson, San Jose, CA, USA) was used to estimate the sub-G1 phase of cells, and the data were evaluated while using BD Accuri™ C6 Software (Version 1.0.264.21, Accuri Cytometers Inc., Ann Arbor, Michigan, United States).

### 4.9. FITC Annexin V Staining

Apoptosis was determined by staining the cells with FITC Annexin V Apoptosis Detection Kit II (Becton-Dickinson Biosciences, CA, USA). The cells were harvested after incubation with UA (0, 10, 25 or 50 µM) 48 h and stained with Annexin V-FITC for 30 min. at 37 °C. The cells analyzed using a FACStar flow cytometer (Becton-Dickinson, San Jose, CA, USA).

### 4.10. siRNA Transfection

The SNU484 and SNU638 cells (10^6^ cells/well) were seeded in a six-well tissue plate. The cells were transfected with 10 μM of RASSF1 siRNA, Yap siRNA or control siRNA (Santa Cruz, CA, USA). The transient transfection was performed while using Lipofectamine 3000 (Invitrogen^TM^, Waltham, MA, USA) reagent according to the manufacturer’s protocol. After 24 h, the SNU484 and SNU638 cells were used for further treatment.

### 4.11. RNA Isolation and Real-Time Polymerase Chain Reaction

After the cells reached 60–80% confluence, they were treated with 0, 25, or 50 µM UA and then incubated for 48 h. Total RNA isolation was performed using TRIzol reagent (Ambion by Life Technologies, Thermo Fisher Scientific Inc., Waltham, MA, USA) following the manufacturer’s protocol. Reverse transcription (PrimeScript^TM^ RT reagent kit, Takara Bio Inc., Otsu, Shiga, Japan) and quantitative real-time polymerase chain reaction (PCR) with SYBR Premix Ex Taq (Takara Bio Inc., Otsu, Shiga, Japan) were performed in an ABI Prism 7900 Sequence Detection System (Applied Biosystems, Foster City, CA, USA). The program was started with 30 s at 95 °C and then 40 cycles of 95 °C for 15 s and of 60 °C for 1 min. The data considered threshold cycle numbers and were normalized with GAPDH. The following primer sequences were used: Rassf1 sense, 5′-CAGATTGCAAGTTCACCTGCCACTA-3′ and antisense 5′-ACCAGCTGCCGTGTGG-3′; YAP sense, 5′-GTGAGGCCACAGGAGTTAGC-3′ and antisense 5′-GGTGCCACTGTTAAGGAAAGG-3′; CTGF sense, 5′-CGACTGGAAGACACGTTTGG-3′ and antisense, 5′-AGGCTTGGAGATTTTGGGAGA-3′; and, GAPDH sense, 5′- GTCTCCTC TGACTTCAACAGCG-3′ and antisense, 5′-ACCACCCTGTT GCTGTAGCCAA-3′.

### 4.12. In vivo Xenograft Animal Study

The animal experiments were performed under the approval of the Institutional Animal Care and Use Committee (IACUC#CBNU2017-0001, 3 January, 2017) of the Chonbuk National University under NIH guidelines (USA). Four-week-old female SPF/VAF immunodeficient mice were obtained from Orient Bio (Dea Jeon, South Korea). The mice were allowed to acclimate to local conditions for one week prior to performing the experiments. Afterwards, 0.1 mL of Matrigel containing 3.5 × 10^6^ human gastric cancer cells (SNU484) was s.c. inoculated into the right flank of the mice. After tumor implantation, the animals were separated into two groups: i) the untreated control group (*n* = 5, DMSO in 50 μL of PBS daily) and ii) the UA-treated group (*n* = 5, 10 mg/kg in 50 μL of PBS once daily). The tumor size was measured once every three days while using a caliper and calculated as (width)^2^ × length/2. The animal experiment was terminated when the tumor was 2 cm in size and body weight was regularly measured before and after UA treatment. The samples were stored at −80 °C.

### 4.13. Histopathological Analysis

The animals were sacrificed with an overdose of ether for 24 h; histologic specimens were obtained from the tumor in the xenograft mice, fixed in 10% formalin, and embedded in paraffin. Serial 5-μM thick sections were cut and stained with H&E.

### 4.14. Statistical Analysis

All of our experiments were repeated three times. The data represented mean values with SE. One-way ANOVA was conducted to evaluate the change among groups. The effects of UA treatments were analyzed with post-hoc tests at the same probability level after ANOVA was performed. A *p*-value < 0.05 was considered to be statistically significant.

## Figures and Tables

**Figure 1 ijms-20-04709-f001:**
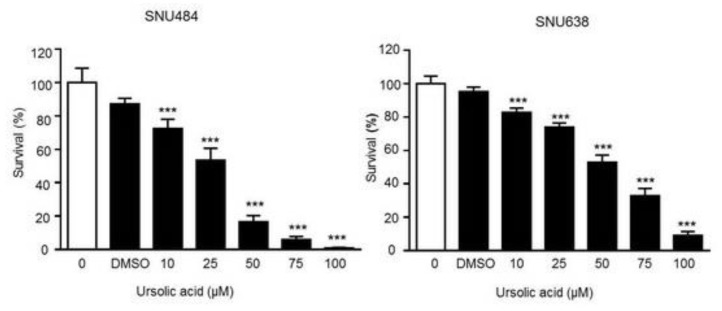
Ursolic acid (UA) inhibited gastric cancer cell growth. Human gastric cancer SNU484 and SNU638 cells were treated with UA (0, 10, 25, 50, 75, and 100 μM) and cell proliferation was detected by MTT assays. Data are mean (SE) of > 3 independent experiments with triplicate dishes. ****p* < 0.01 compared with the control.

**Figure 2 ijms-20-04709-f002:**
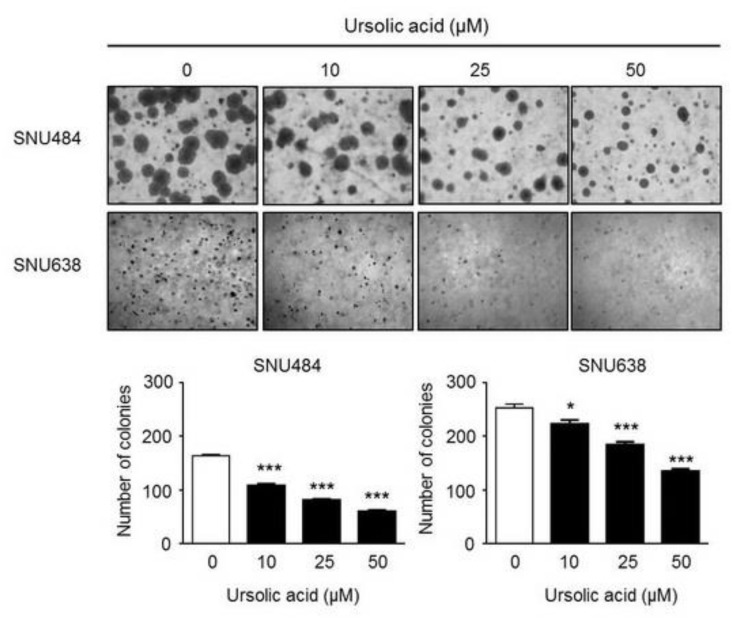
Soft agar colony formation assays. UA significantly the inhibited colony formation of SNU484 and SNU638 cells in a dose-dependent manner (0, 10, 25, 50 μM). Multiplication, 100X. Data represent the mean ± SE of three independent experiments. **p* < 0.05 and ****p* < 0.001 compared to the control.

**Figure 3 ijms-20-04709-f003:**
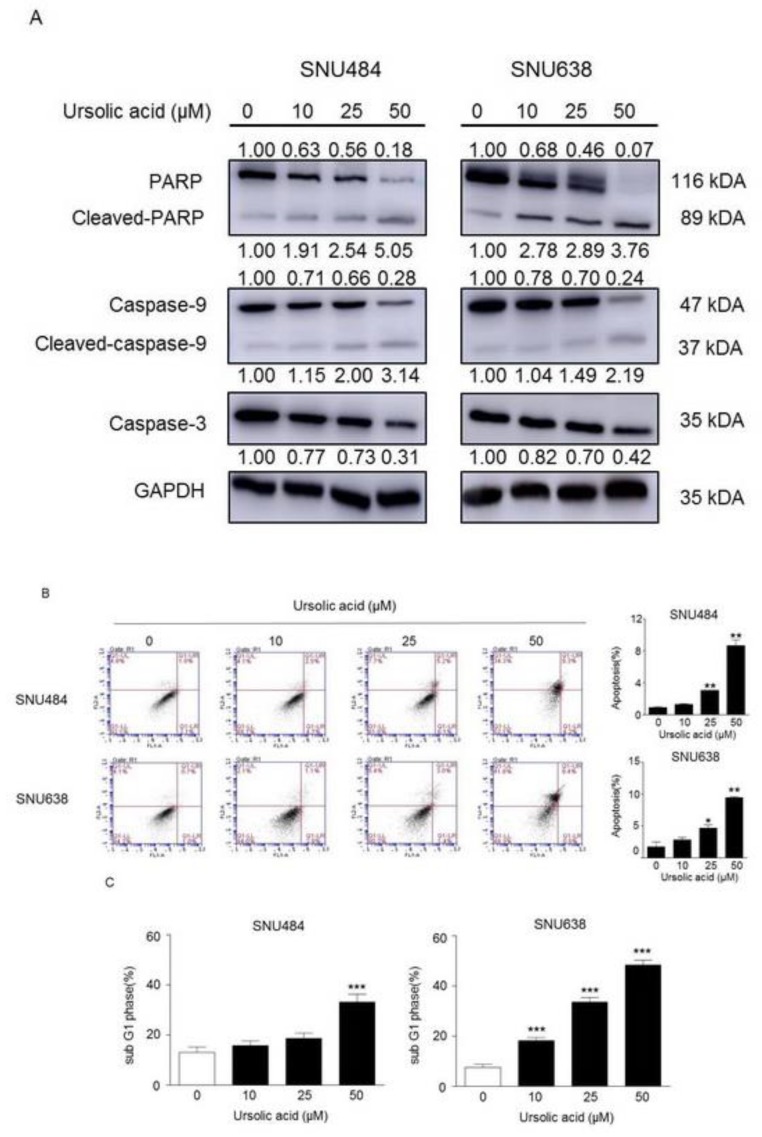
Effects of UA on cell apoptosis. SNU484 and SNU638 cells were treated with UA (0, 10, 25, and 50 μM). (**A**) The apoptosis-regulatory factors cleaved-PARP, PARP, cleaved-caspase-9, caspase-9, and caspase-3 were measured after UA treatment (0, 10, 25, and 50 μΜ) while using Western blot analysis in SNU484 and SNU638 cells. PARP, caspase-9, and caspase-3 were significantly decreased, whereas cleaved-PARP and cleaved-caspase-9 were significantly increased in a dose-dependent manner. GAPDH was used as the internal control. (**B**) The induction of apoptosis was determined through staining with Annexin V/PI and flow cytometry analysis. The apoptotic regions were quantitatively analyzed. (**C**) The cell cycle was detected through staining with PI and flow cytometry analysis. The sub-G1 phase was increased in a dose-dependent manner. Data represent the mean ± SE of three independent experiments. **p*< 0.05, ***p*< 0.01, and ****p* < 0.001 compared with the control.

**Figure 4 ijms-20-04709-f004:**
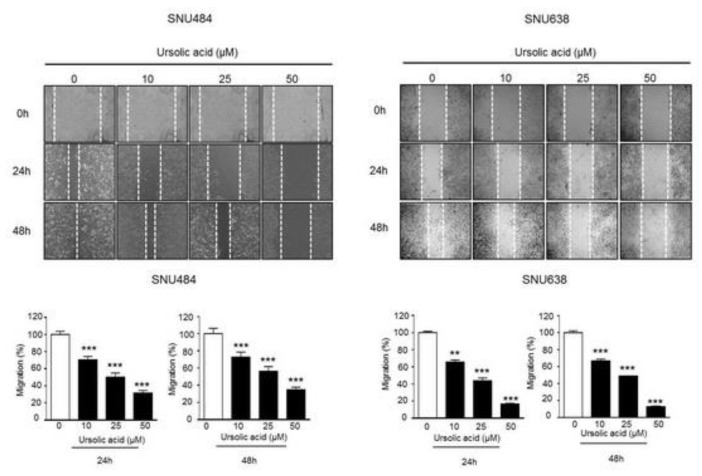
Effect of UA on the migration rates of gastric cancer cells. (**A**) SNU484 and SNU638 cells were treated with UA (0, 10, 25, and 50 μM), the migratory rates were detected by wound healing assay. UA significantly inhibited the cell migration in a dose-dependent manner both at 24 h and 48 h. Data represent the mean ± SE of three independent experiments. ***p* < 0.01 and ****p* < 0.001 compared to the control.

**Figure 5 ijms-20-04709-f005:**
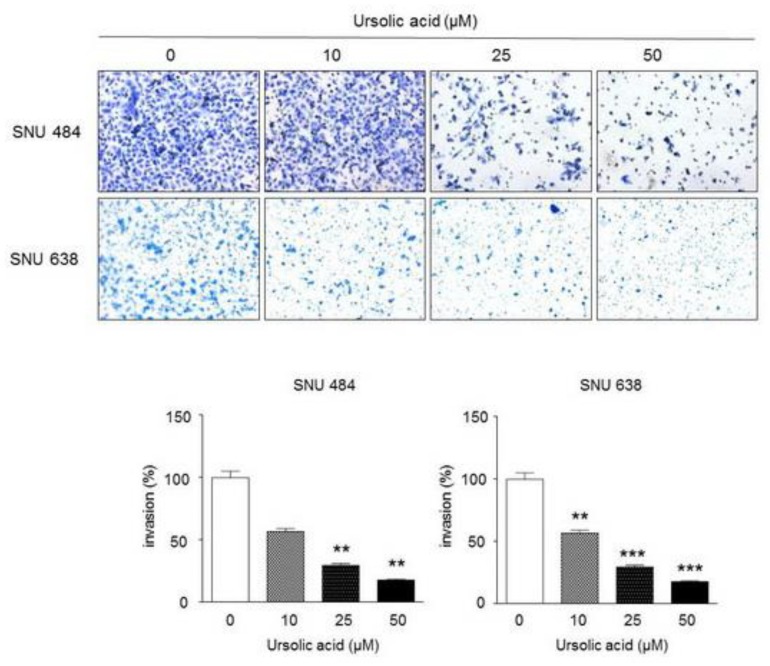
Effect of UA on the invasive rates of gastric cancer cells. SNU484 and SNU638 cells were treated with UA (0, 10, 25, and 50 μM), the invasion rates were detected by Materigel invasion assay. UA significantly inhibited the cell migration in a dose-dependent manner both at 24 h and 48 h. Data represent the mean ± SE of three independent experiments. ***p* < 0.01 and ****p* < 0.001 compared to the control.

**Figure 6 ijms-20-04709-f006:**
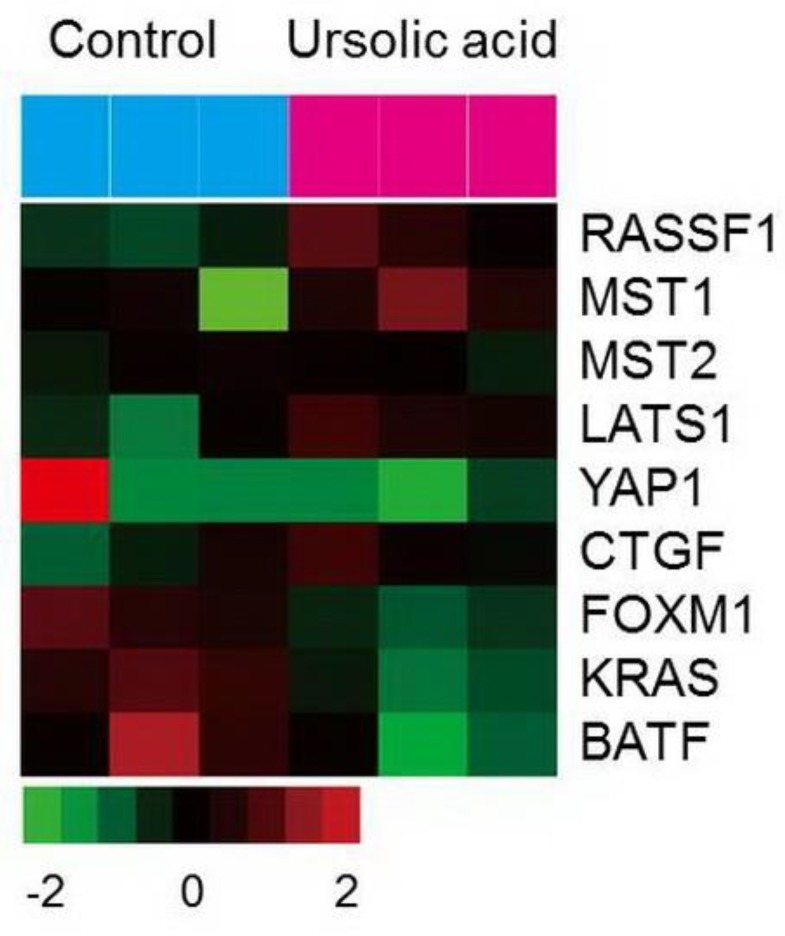
Effect of UA on genes associated with Hippo pathway and oncogenes in SNU484. Microarray heat map is presented in matrix format with rows demonstrating the individual gene and columns demonstrating each samples (control group: *n* = 3; UA-treated group: *n* = 3). Red and green color indicate upregulated and downregulated gene expression levels, respectively, as shown in the scale bar (log 2 transformed). RASSF1; ras association domain-containing protein1, MST1; macrophage-stimulating 1, MST2; macrophage-stimulating 2, LATS1; large tumor suppressor kinase 1, YAP1; yes-associated protein 1, CTGF; connective tissue growth factor, FOXM1; forkhead box protein M1, KRAS; K-ras, BATF; basic leucine zipper transcription factor.

**Figure 7 ijms-20-04709-f007:**
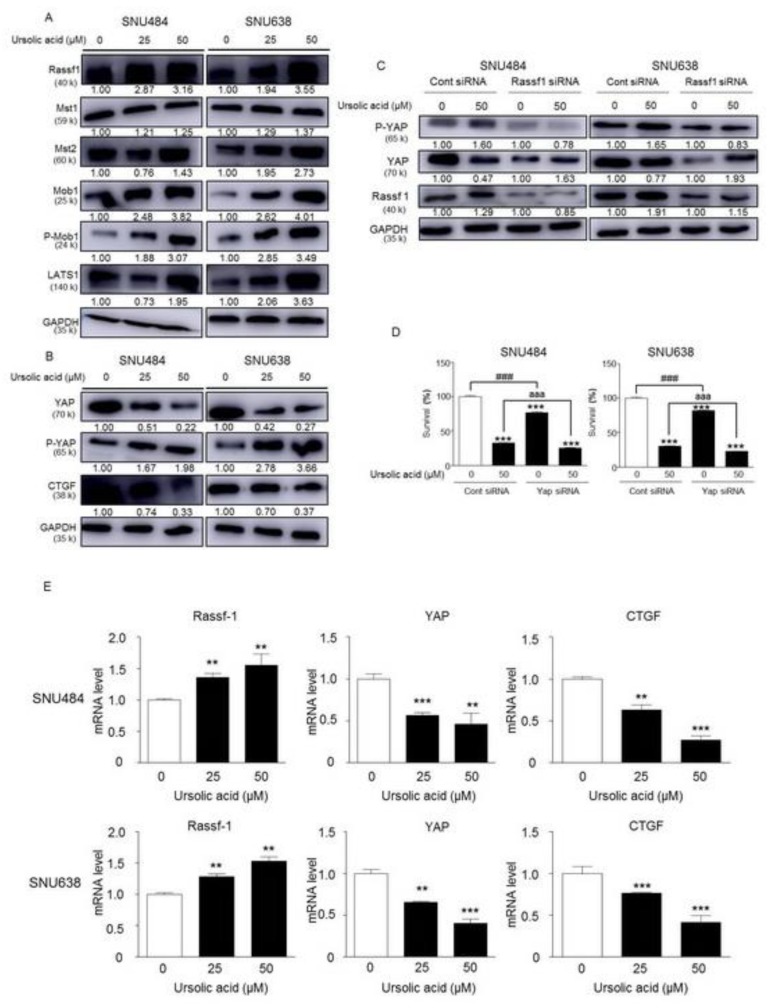
Effect of UA on Hippo pathway-related proteins in SNU484 and SNU638 cells. (**A**) The Hippo pathway-related proteins Rassf1, Mst1, Mst2, Mob1, p-Mob1, and LATS1 were detected after treatment with UA (0, 25, and 50 μM) using Western blot analysis. (**B**) The expressions of CTGF, YAP, and p-YAP proteins treated with UA (0, 25, and 50 μM) in SNU484 and SNU638 cell lines. GAPDH was used as the internal control. (**C**) The effects of RASSF1 inhibition after UA treatment in the Hippo pathway. SNU484 and SNU638 cells were transfected with non-target siRNA in the control group and with RASSF1 siRNA in the experimental group. After 24 h, each group was treated with UA (0 or 50 μM). The expressions of P-YAP, YAP, and RASSF1 were detected while using Western blot analysis. (**D**) The association between Hippo pathway with the cellular phenotypes after treatment of UA. The MTT assay was performed to determine the cell survival after UA treatment in YAP siRNA-transfected SNU484 and SNU638 cells. UA suppressed the cell survivals reduce by Yap silencing in SNU484 and NSU638 cells. ****p* < 0.001 when compared to the control. ^###^*p* < 0.001 when it compared between control siRNA and Yap siRNA treatment. ^aaa^*p* < 0.001 when it compared between control siRNA and Yap siRNA with UA treatment. (**E**) The mRNA levels of Rassf1, YAP, and CTGF were measured through real-time PCR. Data are expressed as mean ± SE. Values were normalized to GAPDH.

**Figure 8 ijms-20-04709-f008:**
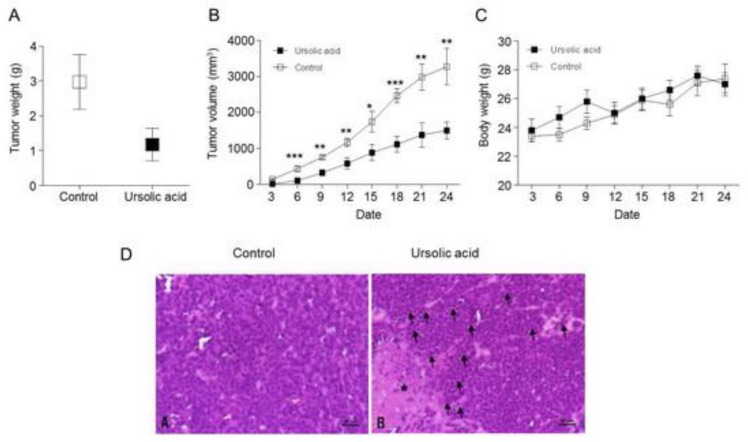
Effects of UA on gastric tumor growth. UA inhibits the growth of gastric tumors of xenograft mice and results are shown for vehicle-treated control and UA-treated group. (**A**) tumor weights were measured after finish of therapy. **p* < 0.05 when compared to the control. (**B**) Tumor volume was measured by a caliper in three times per week and calculated as (width)^2^ × length/2. (UA, ursolic acid), **p* < 0.05, ***p* < 0.01, and ****p* < 0.001 compared with the control at each time point. (**C**) Animal body weights were measured during the course of the experiments. (**D**) Effect of UA on histological tumor tissue. Control group; round to oval hyperchromatic nuclei with scant cytoplasm. UA group; Minimal or mild necrosis (*) but lymphocyte infiltration (black arrows) around tumor cells. hematoxylin and eosin (H&E), bar = 50 μM.

**Figure 9 ijms-20-04709-f009:**
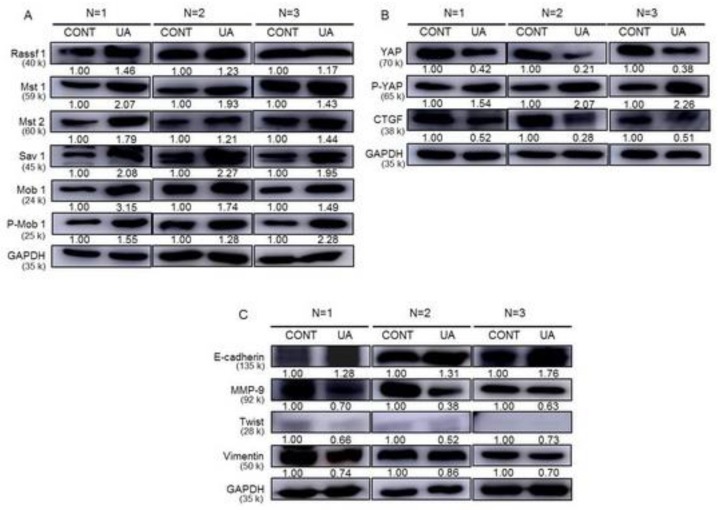
Effects of UA on Hippo pathway-related proteins and metastasis-related proteins in gastric tumors of xenograft mice. (**A**) Western blot data showing the protein expression of Rassf1, Mst1, Mst2, Mob1, p-Mob1, and Sav 1 in tumor tissues from vehicle-treated controls and UA-treated xenografts. (**B**) The expressions of CTGF, YAP, and p-YAP proteins were detected through Western blot analysis in tumor tissues from xenografts. (**C**) The tumor tissues were analyzed for metastasis-related proteins, including E-cadherin, MMP-9, Twist, and Vimentin. GAPDH was used as the internal control. Sav 1, salvador homolog 1; MMP-9, Matrix metallopeptidase 9.

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
