# Peer review of "Activating Hippo Pathway via Rassf1 by Ursolic Acid Suppresses the Tumorigenesis of Gastric Cancer"

_ijms, 2019, doi:10.3390/ijms20194709_

Round 1
Reviewer 1 Report
In this manuscript, Kim and co-authors investigated the effect of Ursolic acid (UA) on the proliferation, apoptosis, migration and invasion ability of gastric cancer cell lines. They found that UA diminishes the proliferation and metastasis of gastric cancer via the regulation of Hippo pathway through Rassf1. The paper is interesting and in good organization.
Specific comments:
Figure 3A, no significant change of Caspase-3 expression is observed as the author claimed. Figure 3B, the axis label and statistics result should be provided. Figure 7C, why the Rassf1 protein level is not changed upon UA treatment since its mRNA level is increased? There is no direct link showing Hippo pathway is responsible for the cellular phenotypes (such as decreased proliferation and increased apoptosis) caused by UA treatment. In Figure 7C, the author should include cellular phenotype assay to demonstrate this. It is hard to interpret the result from Figure 8 and 9, since UA is applied to mouse and no clear conclusion can be acquired. A pretreatment of cells with UA may be helpful.
Author Response
Responses to the Editor’s and Reviewers’ Comments
We appreciate the insightful review of our manuscript by the reviewers of the International Journal of Molecular Science. We have considered each reviewer’s comments carefully and have responded accordingly. Point-by-point responses to the reviewers’ comments and revisions made in the manuscript are listed below. Note that all changes in the manuscript have been marked in red font (additional content).
Ref. No.: IJMS-592553
Title: Activating Hippo pathway via RASSF1 by ursolic acid suppresses tumorigenesis of gastric cancer
Reviewer 1
In this manuscript, Kim and co-authors investigated the effect of Ursolic acid (UA) on the proliferation, apoptosis, migration and invasion ability of gastric cancer cell lines. They found that UA diminishes the proliferation and metastasis of gastric cancer via the regulation of Hippo pathway through Rassf1. The paper is interesting and in good organization.
Specific comments:
Figure 3A, no significant change of Caspase-3 expression is observed as the author claimed.
Response: Your insightful comments are greatly appreciated. We agree with the reviewer that it is essential to confirm the expression of caspase-3 proteins. Accordingly, we performed an experiment to determine the protein levels of caspase-3 bands after treatment of UA (0, 10, 25, 50 μM) in SNU484 cell line. We have revised Fig. 3A in the revised manuscript.
Figure 3B, the axis label and statistics result should be provided.
Response: Your insightful comments are greatly appreciated. As suggested, we measured the percentage levels of the apoptosis after treatment of UA in SNU484 and SNU638 cells by staining with Annexin-V/PI. We found that UA significantly induced apoptosis in a dose-dependent manner in SNU484 and SNU638 cells. We have revised Fig. 3A in the revised manuscript.
Figure 7C, why the Rassf1 protein level is not changed upon UA treatment since its mRNA level is increased? In Figure 7C, the author should include cellular phenotype assay to demonstrate this.
Response: We appreciate your valuable comment. As shown in Fig. 7C, Rassf1 levels were increased after treatment of UA in both protein and mRNA levels in SNU484 and SNU638 cells.
There is no direct link showing Hippo pathway is responsible for the cellular phenotypes (such as decreased proliferation and increased apoptosis) caused by UA treatment.
Response: Your insightful comments are greatly appreciated. We agree with the reviewer that it is essential to confirm the direct link between Hippo pathway with the cellular phenotypes such as cell viabilities. Accordingly, we performed an experiment to determine the cell survival after UA treatment in Yap siRNA-transfected SNU484 and SNU638 cells. As you can see, the cell survival was significantly decreased by YAP silencing. Furthermore, UA further suppressed the cell survivals reduced by Yap silencing in SNU484 and SNU638 cells. These results suggest that there is a direct link showing Hippo pathway is responsible for the cellular phenotype cause by UA treatment. We have included this new information in the revised manuscript (Fig. 7D).
It is hard to interpret the result from Figure 8 and 9, since UA is applied to mouse and no clear conclusion can be acquired. A pretreatment of cells with UA may be helpful.
Response: We appreciate your valuable comments. We used the dose of UA (10 mg/ kg) as that used in a subcutaneous xenograft model from another published article (San et al., 2016). We found that UA treatment resulted in a great suppression of tumor growth in SNU484 xenograft mouse (Fig. 8). In the histological evidence of the xenograft experiment, UA induced minimal necrosis and moderate lymphocyte infiltration around the tumor cells in the UA-treated group. We further harvested the samples after sacrificing the xenograft mice. We further examination to test in vivo UA inhibits the proliferation and metastasis of gastric cancer cells via activation of the Hippo pathway. Therefore, we measured the protein expression levels of Hippo pathway proteins (Rassf-1, Mst1, Mst2, Sav1, Mov1, p-Mob1, Yap, p-Yap, CTGF) and metastasis proteins (E-cadherin, MMP-0, Twist, vimentin) in the control and UA-treated xenograft tumor tissues. We found that the protein levels of Rassf-1, Mst1, Mst2, Sav1, Mov1 and p-Mob1 were increased in the UA-treated xenograft tumor tissues compared to those of the control tumor tissues. The protein levels of Yap and CTGF were decreased in the UA-treated xenograft tumor tissues compared to the control tumor tissues. In addition, the E-cadherin protein level was increased whereas the MMP9, Twist, and vimentin protein levels were decreased in the UA-treated xenograft tumor tissues compared to the control tumor tissues (Fig. 9). These data suggest that UA inhibits the proliferation and metastasis of gastric cancer cells via activation of the hippo pathway in vivo and in vitro.
Reference
Shan, J et al., Ursolic acid synergistically enhances the therapeutic effects of oxaliplantin in colorectal cancer. Protein Cell, 7(8):571-585, (2016)

Reviewer 2 Report
The manuscript is a comprehensive study that includes full evaluation of the cytotoxicity properties of ursolic acid across several cell lines with the proper controls. They evaluated cell viability, apoptosis, cell cycle, tumorogenesis potential/migration/invasion and cell signling analysis. Further validation was carried out in vivo and they provide gene analysis supporting the claim that the hippo pathway is being targeted by the compound. The data is scientific solid and it is a good timing. There is great interest in the field for the Hippo signaling pathway in various cancer models.
please check spelling and references. For instance the word annexin is misspelled.
Author Response
Reviewer 2
The manuscript is a comprehensive study that includes full evaluation of the cytotoxicity properties of ursolic acid across several cell lines with the proper controls. They evaluated cell viability, apoptosis, cell cycle, tumorigenesis potential/migration/invasion and cell signaling analysis. Further validation was carried out in vivo and they provide gene analysis supporting the claim that the hippo pathway is being targeted by the compound. The data is scientific solid and it is a good timing. There is great interest in the field for the Hippo signaling pathway in various cancer models.
please check spelling and references. For instance, the word annexin is misspelled.
Response: Thank you very much for the valuable comments. We have checked spelling and references in the revised manuscript.
Round 2
Reviewer 1 Report
All my concerns are addressed properly.